# Effects of Hip Structure Analysis Variables on Hip Fracture: A Propensity Score Matching Study

**DOI:** 10.3390/jcm8101507

**Published:** 2019-09-20

**Authors:** Yong-Chan Ha, Jun-Il Yoo, Jeongkyun Yoo, Ki Soo Park

**Affiliations:** 1Department of Orthopaedic Surgery, Chung-Ang University College of Medicine, Seoul 06974, Korea; hayongch@naver.com; 2Department of Orthopaedic Surgery, Gyeongsang National University Hospital, Jinju 52727, Korea; 3Department of Orthopaedic Surgery, Gyeongsang National University College of Medicine, Jinju 52727, Korea; luxiant@naver.com; 4Institute of Health Sciences, Gyeongsang National University, Jinju 52727, Korea; 5Department of Preventive medicine, Gyeongsang National University School of Medicine, Jinju 52727, Korea

**Keywords:** elderly, hip fracture, sarcopenia, hip structure analysis

## Abstract

The purpose of this retrospective study was to compare the hip structural analysis (HSA) levels of patients with those of a hip fracture group. All patients with an initial hip fracture who were older than or equal to 65 years old and admitted to our hospital between March 2018 and January 2019 were eligible for this study. During the study period, 134 hip fracture patients aged 65 years and older were admitted to the study institution, and a total of 51 hip fracture patients were ultimately assigned to the patient group. Age, sex, body mass index (BMI), skeletal muscle index (SMI), and vitamin D were matched in the two groups (hip fracture (HF) group vs. non-hip fracture group) using propensity score matching (PSM) without any statistical differences. Following propensity score matching, 51 patients in the HF group and 51 patients in the non-HF group were included in the study, respectively. Hip axis length (*p* = 0.031), neck-shaft angle (*p* = 0.043), width of intertrochanter (*p* = 0.005), and femur shaft (*p* = 0.01) were found to be significantly higher in the HF group (107.31 (mean) ± 9.55 (standard deviation, SD), 131.11 ± 5.29, 5.57 ± 0.58, and 3.05 ± 0.23, respectively) than in the non-HF group (102.07 ± 14.15, 128.85 ± 5.81, 5.29 ± 0.38, and 2.92 ± 0.23, respectively). However, cross-sectional area (CSA) of femur neck (*p* = 0.005) and femur shaft (*p* = 0.01) as well as cortical thickness (CT) of femur neck (*p* = 0.031) and femur shaft (*p* = 0.031) were found to be significantly lower in the HF group (1.93 ± 0.44, 3.18 ± 0.83, 0.11 ± 0.02, and 0.38 ± 0.09, respectively) than in the non-HF group (2.12 ± 0.46, 3.57 ± 0.78, 0.13 ± 0.03, and 0.47 ± 0.11, respectively). The HSA showed excellent sensitivity (82.4% to 90.2%). HSA is an important factor in predicting the occurrence of hip fracture. Therefore, not only should bone mineral density (BMD) be considered clinically, but it is also important to look closely at HSA for risk of hip fracture.

## 1. Introduction

As the global population ages, the life expectancy among the elderly will continue to increase [1]. With the increasing prevalence of various chronic diseases, incidence of osteoporosis-caused fragility fractures is on the rise in the elderly [2]. In particular, among the osteoporotic fractures, hip fractures greatly increase morbidity and mortality rates among the elderly, thereby significantly reducing their qualities of life for the rest of their lives [3,4,5].

Therefore, the prediction and prevention of osteoporotic fractures are crucial, and they can reduce socioeconomic burden [6]. FRAX is the most widely-used fracture prediction program worldwide [7]. In addition, bone mineral density (BMD) measurement using dual-energy X-ray absorptiometry (DXA) equipment is the most widely-used diagnostic tool for osteoporosis, and along with having been used for a long time, is the most widely-used method for predicting fractures and evaluating prognosis [8]. Recently, various measurement values have been developed and used in clinical practice by using a BMD-measuring device. Some of the representative methods are trabecular bone score (TBS), hip structural analysis (HSA), and body composition including skeletal muscle index (SMI) [9,10,11,12,13].

In real clinical practice, only BMD is used for fracture prediction and osteoporosis treatment. However, actual BMD is often higher in elderly patients due to osteoarthritis of the spine and implant surgery [14,15,16]. In addition, most of the fragile fractures occur in patients with osteopenia [17]. For these reasons, recent studies have been conducted to predict fractures using figures such as HSA and TBS with BMD [18]. Based on results from these studies, the International Society of Clinical Densitometry (ISCD) published a position statement in 2015 that with the exception of hip axis length, hip structural geometry parameters should not be used to assess risk of hip fracture [18]. However, previous ISCD position statements for HSA did not reflect on skeletal muscle mass, vitamin D, and other major risk factors for hip fractures. In addition, many studies still report that the thickness and geometry of the cortical bones around the hips are important risk factors for hip fractures. Therefore, we matched risk factors including skeletal muscle mass and vitamin D by using the propensity score matching (PSM) method and analyzed the effect of HSA on hip fracture.

The purpose of this retrospective study was to compare the hip structural analysis (HSA) levels of patients with those of a hip fracture group.

## 2. Materials and Methods

### 2.1. Ethics Statement

The design and protocol of this retrospective study were approved by the Institutional Review Board of our hospital (GNUH-2017-06-008-003). Informed consent was waived by the board.

Data from the 2008 Korean National Health and Nutrition Examination Survey (KNHANES), bearing the approval number of 2008–04EXP-01-C, were reviewed and approved by the Institutional Review Board of the Korea Centers for Disease Control and Prevention (KCDC). Informed consent was obtained from all participants when the 2008 KNHANES was conducted.

### 2.2. Participants

All patients with a hip fracture who were at least 65 years old and admitted to our hospital between March 2018 and January 2019 were eligible for this study. During the study period, 134 hip fracture patients aged 65 years and older were admitted to the study institution. Of these, 35 (26.1%) were excluded because there was no time to perform DXA preoperatively due to the need for urgent surgical repair, 15 (11.2%) were excluded because they refused examination, and 33 (24.6%) were excluded due to mental health issues such as dementia, delirium, depression, and mental retardation. In total, 51 hip fracture patients were ultimately assigned to the patient group.

The KNHANES for the Republic of Korea (ROK) population is a nationwide representative cross-sectional survey with a clustered, multistage, stratified, and rolling sampling design. The KNHANES consists of a health interview, health examination, and dietary survey. The survey data are collected from household interviews and direct standardized physical examinations conducted in specially-equipped mobile examination centers [19]. The data were collected in 2008 from 9744 participants. Patients under 65 years of age who were missing data regarding skeletal muscle mass were excluded. Following these exclusions, 1050 participants were ultimately analyzed. Age, sex, body mass index (BMI), skeletal muscle index (SMI), vitamin D, and osteoporosis status were matched in the two groups (hip fracture (HF) group vs. non-HF group) using propensity score matching (PSM) without a statistical difference (Figure 1).

### 2.3. Biochemical Analyses

The serum 25-hydroxyvitamin D (25(OH) vitamin D) level was measured using the 1470 Wizard gamma counter (Perkin Elmer, Turku, Finland), automatic analyzer 7600 (Hitachi, Tokyo, Japan), and LIAISON (DiaSorin, Stillwater, OK, USA) with a radioimmunoassay (25-hydroxyvitamin D ^125^ I RIA Kit; DiaSorin).

### 2.4. Measurements of the Appendicular Skeletal Muscle Mass and Bone Mineral Density (BMD)

Body composition and bone mineral density (BMD) were measured through whole-body dual X-ray absorptiometry (DXA) using the QDR 4500A apparatus (Hologic, Long Island City, NY, USA) in both the groups (HF and non-HF group). The bone mineral content, fat mass, and lean soft-tissue mass were measured separately for each part of the body, including the arms and legs. It should be noted that the lean soft-tissue masses of the arms and legs are nearly equal to the skeletal muscle mass. Due to the fact that absolute muscle mass is correlated with height, the skeletal muscle index (SMI) was calculated as follows: Lean mass (kg)/height (m^2^). The arm SMI is defined as the arm lean mass (kg)/height (m^2^), while the leg SMI is defined as the leg lean mass (kg)/height (m^2^). The appendicular SMI is defined as the sum of the arm and leg SMIs.

### 2.5. Definition of Osteoporosis

Osteoporosis was defined as BMD 2.5 standard deviations (SDs) below the peak bone mass of a young, healthy, gender- and race-matched reference population, according to the World Health Organization (WHO) diagnostic classification. T-score was used to classify osteoporosis (T-score ≤ −2.5), osteopenia (−2.5 < T-score < −1.0), and normal (T-score ≥ −1) [20].

### 2.6. Hip Structure Analysis (HSA)

In order to evaluate the hip bone geometry, DXA scans were analyzed at the femoral neck (FN), intertrochanteric region (IT), and femoral shaft (FS) using the HSA program. The cross-sectional area (CSA), width (WD), and cortical thickness (CT) were measured based on the bone mass profiles. The hip axis length (HAL) was measured along the femoral neck axis from the base of the greater trochanter to the inner pelvic brim. We calculated the neck shaft angle (NSA) as the angle between the neckline and a line through the shaft of the femur, which were all set by the Hologic software on the outer cortex of the femoral shaft below the region of interest [21,22,23].

### 2.7. Statistical Analyses

For the selection of the control group, a propensity-score matching method was used. The factors considered to be the most important confounders affecting the occurrence of hip fracture were chosen for the propensity score algorithm. A logistic model with hip fracture as the outcome and age, sex, BMI, SMI, vitamin D, and osteoporosis status as confounders were used to estimate the propensity score. We then matched each conversion arthroplasty patient to a control subject based on the propensity score. The maximum difference between propensity probabilities for matching was set at 0.1. To account for the matched design, we performed paired t-tests [24].

In order to compare the means and proportions of each group, the Student’s t-test and the chi-squared (χ^2^) test were conducted. Variables with *p*-values of <0.05 were included in the multivariate model. The associations between the clinical characteristics and bone variables were given as the Pearson correlation coefficients. A receiver operator curve analysis (ROC) was also performed to identify the cut-off value for diagnosis of hip fracture using hip structural analysis. All of the statistical tests were two-tailed, and statistical significance was defined as *p* < 0.05. All of the statistical calculations were performed using SPSS Statistics V.22 (SPSS, NY, USA).

## 3. Results

### 3.1. Demographic Characteristics by Presence of Hip Fracture after Propensity Score Matching

Following propensity score matching, 51 patients in the HF group and 51 patients in the non-HF group were ultimately included in the study, respectively. In the comparison of the demographic data by the presence of hip fracture, age (*p* = 0.807), sex (*p* = 1.00), BMI (*p* = 0.5), osteoporosis (*p* = 0.89), SMI (*p* = 0.429), and vitamin D (*p* = 0.934) showed no statistically significant differences between the groups (Table 1).

### 3.2. Hip Structural Analysis (HSA) by the Presence of Hip Fracture

HAL (*p* = 0.031), NSA (*p* = 0.043), and WD of intertrochanter (*p* = 0.005) and femur shaft (*p* = 0.01) were found to be significantly higher in the HF patients (107.31 ± 9.55, 131.11 ± 5.29, 5.57± 0.58, and 3.05 ± 0.23, respectively) than in the non-HF patients (102.07 ± 14.15, 128.85 ± 5.81, 5.29 ± 0.38, and 2.92 ± 0.23, respectively).

However, CSA of femur neck (*p* = 0.005) and femur shaft (*p* = 0.01) as well as CT of femur neck (*p* = 0.031) and femur shaft (*p* = 0.031) were found to be significantly lower in the HF patients (1.93 ± 0.44, 3.18 ± 0.83, 0.11 ± 0.02, and 0.38 ± 0.09, respectively) than in the non-HF patients (2.12 ± 0.46, 3.57 ± 0.78, 0.13 ± 0.03, and 0.47 ± 0.11, respectively) (Table 2).

### 3.3. Receiver Operator Curve (ROC) Analysis for Diagnosis of Hip Fracture Using Hip Structural Analysis

The results of ROC analysis showed that the cut-off points of hip structural analysis were 98.73 in HAL (area under the curve (AUC) = 0.587, *p* < 0.001), 2.283 in CSA of femur neck (AUC = 0.617, *p* < 0.001), 5.712 in WD of intertrochanter (AUC = 0.637, *p* < 0.001), 3.738 in CSA of femur shaft (AUC = 0.653, *p* < 0.001), 2.801 in WD of femur shaft (AUC = 0.5621, *p* < 0.001), and 126.4 in NSA (AUC = 0.604, *p* < 0.001) (Figure 2). The sensitivity and specificity are shown in Table 3. The hip structural analysis showed excellent sensitivity (82.4% to 90.2%).

## 4. Discussion

The main finding of this study was that HSA affects the occurrence of hip fractures. We analyzed the effects on HSA after ensuring that both the groups were statistically similar through PSM. In addition, ROC analysis was performed to determine the cut-off point in order to predict the occurrence of hip fracture for each HSA factor.

Kaptoge et al. tested whether section modulus, a geometric index of bending strength, can predict hip fracture better than BMD. The target variables were the narrow neck (NN), intertrochanter (IT), and shaft (S) regions. By analyzing the IT region, Kaptoge et al. best predicted trochanteric fracture risks [25].

Khoo et al. used two-dimensional DXA to predict the fracture risk of proximal femur and applied the same measure to three proximal femur sites—the femoral neck, intertrochanter, and shafts. In total, twelve variables were used in the study: Each of the three proximal femur sites with the localized areal bone mineral density (aBMD) and the sub-periosteal width, standard deviation of the normalized mineral mass projection profile distribution, and displacement between center of mineral mass and geometric center of mineral mass of the projection profile. This study showed that the addition of sigma at the intertrochanter site, which is a combination of intertrochanter and standard deviation of normalized mineral mass projection, to total hip aBMD and age significantly improved the prediction of 15-year hip fracture probability in elderly women (with a mean baseline age of 75 [25].

Leslie et al. questions the types of bone geometric measures that can best explain the fracture prediction independent of conventional BMD. Among women aged over 50, hip axis length (HAL) and strength index (SI) showed statistically significant contributions to hip fracture prediction independent of age and hip BMD measurement. HAL and SI well predict hip fracture risks, but not non-hip fracture risks [26].

Among postmenopausal women, LaCroix et al. studied the intertrochanter outer diameter and buckling ratio, which are the two hip geometry parameters that best predict incidence of hip fracture at the intertrochanter and at the shaft. Of the HSA parameters, bone cross-sectional area, outer diameter, selection modulus, and buckling ratio were applied to three hip regions—narrow neck, intertrochanter, and shaft. In addition, aBMD was used as a comparison model [27,28].

Leslie et al. studied 13,978 subjects aged ≥50 years and 268 subjects with hip fracture over six years. The finite element analysis (FEA) method involves considering the additional geometric properties that impact bone strength and fracture susceptibility. Automatically-derived femoral neck, intertrochanter, and subtrochanter fracture risk indices were associated with the incidence of hip fracture independent of multiple covariates, including femur neck BMD, fracture risk assessment probability and risk factors, and hip axis length [29].

Aldieri et al. studied 28 post-menopausal female subjects aged 55–81 years. The two-dimensional simplified FE analyses performance may be integrated in DXA devices in the future. HSA-based hip fracture risk estimator could support the current standard, which is BMD [30].

Borggrefe et al. evaluated quantitative computed tomography (QCT) scans of 230 men (65 subjects with hip fracture). Age, BMI, and site-adjusted hazard ratios were found to be significant for all measures and additionally included aBMD buckling ratio, and trabecular volumetric BMD (vBMD) remained associated with hip fracture risk. QCT-based studies help predict fractures, but their results are not as strong [31].

Fujii et al. studied 49 patients with diabetes mellitus, and BMD at DXA and tomosynthesis image indices were compared between the vertebral compression fracture group of 16 patients and non-fracture group of 33 patients. The averages of bone volume per tissue volume and entropy at the principal tensile group in the vertebral compression fracture group were lower than those in the non-fracture group. Tomosynthesis-based trabecular bone analysis is technically feasible and combined with BMD measurements, can potentially be used to determine bone strength in patients with diabetes mellitus [32].

In a study conducted by Kolta et al., trabecular bone score (TBS) of the lumbar spine was assessed in 122 patients aged ≥50 years on a low-dose imaging system (EOS). TBS-EOS was found to be lower in patients with severe low-trauma fractures than non-fractured patients independently from BMD. The area under ROC curve for TBS-EOS was 0.70 [33].

Yang et al. investigated 324 patients from a total of 982 aged ≥65 years who had been identified with hip fracture. Fracture risk indices (FRIs) derived from DXA-based FEA were independently associated with prior hip fracture. When stratified by hip fracture probability, intertrochanteric and subtrochanteric FRIs were more strongly associated with hip fracture in women with hip fracture probability <3% than in those with hip fracture probability ≥3% [34]. In a previous study, Faulkner and McClung et al. [35] reported that an automatic hip axis length measurement of 11.0 cm is associated with a two-fold increase in hip fracture risk compared with a woman with an average hip axis length. In addition, they suggested that a hip axis length value of 11.6 cm increases hip fracture risk by a factor of 4 compared with a woman with a normal hip dimension. In this study, the same cut-off points were not calculated because different races were analyzed in previous studies. However, the trend of increasing HAL in the fracture group was the same pattern as the previous study. Therefore, the cut-off value measured in this study will need to be validated on a larger sample cohort.

There are several limitations to this study. First, the study sample was small. However, statistical problems were solved by comparing the two groups through propensity score matching. Second, the research design was retrospective. Therefore, it is necessary to carry out research specifically designed for this purpose. Third, the time difference between the patient and control groups was measured. However, the DXA equipment used in this patient control is the same machine as Hologic equipment. In addition, the BMD measurement protocol used in the control group was used in the patient group. Fourth, we could not classify all types of hip fracture due to the limited number of samples in this study. Hip fractures include femur neck and intertrochanter, which can cause variability when interpreting the HSA values of the femoral region. In future studies, it should be considered that factors such as fracture type, treatment for osteoporosis, and duration of treatment can affect the thickness and width of the cortical bone around the femur. Fifth, the historical data with time gap was used as a control. However, the hip structural analysis data for a large number of healthy people were measured in the KNHANES study only in 2008, so these data were used in this study. Since there is still debate on the usefulness of hip structural analysis, this preliminary study was conducted to obtain additional evidence. Therefore, further studies are needed in larger sample cohorts. Finally, matching for all comorbidities could not be performed due to limitations in retrospective data analysis. Among the sociodemographic variables, economic and educational levels were different from those of the control group’s national data-based survey. Therefore, we could not perform statistical analysis after matching socio-economic variables. In disease and laboratory tests, osteoporosis, skeletal muscle mass, and vitamin D were matched and used as propensity variables.

In conclusion, hip structural analysis is an important factor in predicting the occurrence of hip fracture. Therefore, not only should BMD be considered clinically, but it is also important to look closely at HSA for risk of hip fracture.

## Figures and Tables

**Figure 1 jcm-08-01507-f001:**
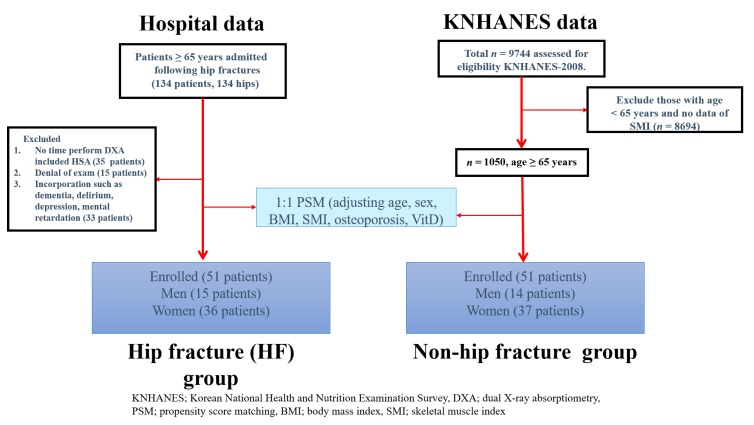
Flow chart of this study. KNHANES, Korean National Health and Nutrition Examination Survey; HSA, hip structure analysis; DXA, dual X-ray absorptiometry; PSM, propensity score matching; BMI, body mass index; SMI, skeletal muscle index; VitD, vitamin D.

**Figure 2 jcm-08-01507-f002:**
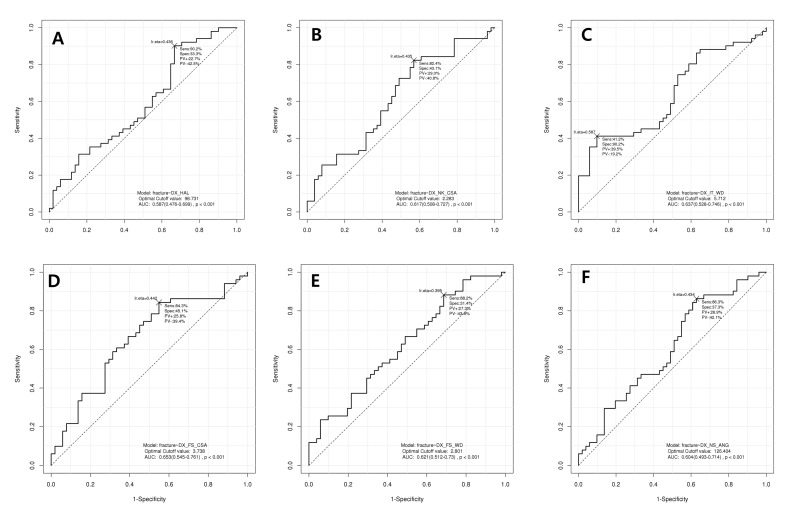
Receiver operating curve (ROC) analysis for diagnosis of hip fracture using hip structural analysis. (**A**) Hip axis length (HAL), (**B**) cross-sectional area (CSA) of femur neck, (**C**) width (WD) of intertrochanter, (**D**) cross-sectional area (CSA) of femur shaft, (**E**) width (WD) of femur shaft, and (**F**) neck-shaft angle (NSA).

**Table 1 jcm-08-01507-t001:** Demographic characteristics by presence of hip fracture after propensity score matching.

	Non-HF (*N* = 51)	HF (*N* = 51)	*p*-Value
Age (years)	78.2 ± 6.4	77.9 ± 7.3	0.807
Sex			1.000
Male	14 (27.5%)	15 (29.4%)	
Female	37 (72.5%)	36 (70.6%)	
BMI (kg/m^2^)	22.0 ± 3.0	21.6 ± 3.4	0.500
Osteoporosis			0.890
Normal	3 (5.9%)	2 (3.9%)	
Osteopenia	15 (29.4%)	16 (31.4%)	
Osteoporosis	33 (64.7%)	33 (64.7%)	
SMI (kg/m^2^)	6.0 ± 0.8	5.8 ± 1.0	0.429
VitD (ng/mL)	17.5 ± 9.0	5.8 ± 1.0	0.934

HF, hip fracture; BMI, body mass index; SMI, skeletal muscle index; VitD, vitamin D.

**Table 2 jcm-08-01507-t002:** Hip structural analysis (HSA) by the presence of hip fracture.

		Non-HF (*N* = 51)	HF (*N* = 51)	*p*-Value
**Hip axis length (mm)**		102.07 ± 14.15	107.31 ± 9.55	0.031
Femur neck	CSA	2.12 ± 0.46	1.93 ± 0.44	0.030
	WD	3.37 ± 0.33	3.41 ± 0.34	0.606
	CT	0.13 ± 0.03	0.11 ± 0.02	0.004
Intertrochanteric area	CSA	3.28 ± 0.88	3.18 ± 0.88	0.535
	WD	5.29 ± 0.38	5.57± 0.58	0.005
	CT	0.26 ± 0.07	0.24 ± 0.06	0.076
Femur shaft	CSA	3.57 ± 0.78	3.18 ± 0.83	0.016
	WD	2.92 ± 0.23	3.05 ± 0.23	0.010
	CT	0.47 ± 0.11	0.38 ± 0.09	<0.001
NSA (°)		128.85 ± 5.81	131.11 ± 5.29	0.043

HF, hip fracture; CSA, cross-sectional area; WD, width; CT, cortical thickness; NSA, neck-shaft angle.

**Table 3 jcm-08-01507-t003:** ROC analysis for diagnosis of hip fracture using hip structural analysis.

HSA	Cut-Off Point	Sensitivity	Specificity	AUC	*p*-Value
HAL	98.73	90.2%	33.3%	0.587	<0.001
NK_CSA	2.283	82.4%	43.1%	0.617	<0.001
IT_WD	5.712	41.2%	90.2%	0.637	<0.001
FS_CSA	3.738	84.3%	45.1%	0.653	<0.001
FS_WD	2.801	88.2%	31.4%	0.621	<0.001
NSA	126.40	86.3%	37.3%	0.604	<0.001

ROC, receiver operating curve; AUC, area under the ROC curve; HSA, hip structural analysis; HAL, hip axis length; NK_CSA, cross-sectional area of femur neck; IT_WD, width of intertrochanteric area; FS_CSA, cross-sectional area of femur shaft; FS_WD, width of femur shaft; and NSA, neck-shaft angle.

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
