# Peer review of "Effects of Hip Structure Analysis Variables on Hip Fracture: A Propensity Score Matching Study"

_jcm, 2019, doi:10.3390/jcm8101507_

Round 1

Reviewer 1 Report

This paper compared hip structural geometry measures among older adults with versus without a hip fracture using propensity score matching on key variables, and identified clinical cut points for these measures for diagnosing hip fracture. While this is an interesting analysis, it is unclear what this adds to the literature. There have been quite a few studies that have evaluated the relationship between bone geometry and hip fracture. Based on results from these studies, the International Society of Clinical Densitometry published a position statement in 2015 that with the exception of hip axis length, hip structural geometry parameters should not be used to assess risk of hip fracture. I recommend re-framing the paper to address how this analysis is a unique contribution, particularly in light of the ISCD statement. Here are my specific comments:

Consider adding sub-headings to describe the hip fracture and non-hip fracture cohorts and spell out all abbreviations. It was unclear in the text that the hip fracture patients were drawn from a prospective sample and the non hip fracture cohort came from a previous study. In the fracture cohort, were these all incident fractures? In other words, did you exclude patients with a history of fracture? You should indicate in the methods section that the same scanner was used to measure BMD and skeletal muscle mass in both cohorts. Was the SMI used in the study for total body or appendicular only? Did you also match on osteoporosis status? It is not clear from the text whether this was done. Was consideration given to matching on residential status (were these community living adults? Or were some living in an institution?); and did you exclude from the control group anyone with a mental disorder and dementia? The statistical analysis section does not mention the ROC analysis. When comparing HSA values between the groups, consider using paired t tests to account for the matched design (see Austin 2007; A critical appraisal of propensity-score matching in the medical literature between 1996-2003). In the discussion, address the variability in results by femur location (i.e. neck, shaft and IT). When predicting fracture, is one more important than the other or should all three regions be taken into consideration? There was no text in the discussion comparing these values to other published data- would be nice to see whether they are similar – especially the hip axis length. I think that a next step worth mentioning is that these cut points should be tested in future studies to determine if they are as effective as BMD in predicting fracture. Also address that the current analysis pooled data from men and women when there may be important differences worth investigating in larger samples.

Author Response

Reviewer 1

Comments and Suggestions for Authors

This paper compared hip structural geometry measures among older adults with versus without a hip fracture using propensity score matching on key variables, and identified clinical cut points for these measures for diagnosing hip fracture. While this is an interesting analysis, it is unclear what this adds to the literature. There have been quite a few studies that have evaluated the relationship between bone geometry and hip fracture. Based on results from these studies, the International Society of Clinical Densitometry published a position statement in 2015 that with the exception of hip axis length, hip structural geometry parameters should not be used to assess risk of hip fracture. I recommend re-framing the paper to address how this analysis is a unique contribution, particularly in light of the ISCD statement. Here are my specific comments:

 Response: Thank you for your valuable comments. We have described “Based on results from these studies, the International Society of Clinical Densitometry (ISCD) published a position statement in 2015 that with the exception of hip axis length, hip structural geometry parameters should not be used to assess risk of hip fracture. [18]  However, previous ISCD position statements for HSA did not correct skeletal muscle mass, vitamin D, and other major risk factors for hip fractures. In addition, many studies still report that the thickness and geometry of the cortical bones around the hips are important risk factors for hip fractures. Therefore, we matched risk factors including skeletal muscle mass and vitamin D using PSM method and analyzed the effect of HSA on hip fracture.” in the Introduction.

Consider adding sub-headings to describe the hip fracture and non-hip fracture cohorts and spell out all abbreviations.

 Response: According to your comment, we have added subheadings in manuscripts.

It was unclear in the text that the hip fracture patients were drawn from a prospective sample and the non hip fracture cohort came from a previous study. In the fracture cohort, were these all incident fractures? In other words, did you exclude patients with a history of fracture?

Response: We have already described “The design and protocol of this retrospective study were approved by the Institutional Review Board of our hospital (GNUH-2017-06-008-003).”

To avoid mis-understanding, we have changed “All patients with a fresh hip fracture who were at least 65 years old and admitted to our hospital between March 2018 and January 2019 were eligible for this study.” to “All patients with a hip fracture who were at least 65 years old and admitted to our hospital between March 2018 and January 2019 were eligible for this study.in the Methods.

You should indicate in the methods section that the same scanner was used to measure BMD and skeletal muscle mass in both cohorts.

Response: We have changed “Body composition and bone mineral density (BMD) were measured through whole-body dual X-ray absorptiometry (DXA) using the QDR 4500A apparatus (Hologic, U.S.A.).” to “Body composition and bone mineral density (BMD) were measured through whole-body dual X-ray absorptiometry (DXA) using the QDR 4500A apparatus (Hologic, U.S.A.) in both group (HF and non-HF group). in the Methods.

Was the SMI used in the study for total body or appendicular only? Did you also match on osteoporosis status? It is not clear from the text whether this was done.

Response: We have used appendicular SMI in this study. As shown in Table 1, we matched on the status of osteoporosis. Therefore, we have changed “Age, sex, body mass index (BMI), skeletal muscle index (SMI), and vit D were matched in the two groups (hip fracture patient vs. control group) using propensity score matching (PSM) without a statistical difference (Fig 1).” to “Age, sex, body mass index (BMI), skeletal muscle index (SMI), vit D, and osteoporosis status were matched in the two groups (hip fracture (HF) group vs. non-HF group) using propensity score matching (PSM) without a statistical difference (Fig 1).” in the Methods.

Was consideration given to matching on residential status (were these community living adults?

Or were some living in an institution?); and did you exclude from the control group anyone with a mental disorder and dementia?

Response: We absolutely agree with your comment. We have described “Finally, adjustment for all comorbidities could not be performed due to limitations in retrospective data analysis. Among the sociodemographic variables, economic and educational levels were different from those of the control group's national data-based survey. Therefore, direct statistical analysis could not be performed. However, gender and age were corrected for demographic variables that had the greatest impact on fractures. In disease and laboratory tests, osteoporosis, skeletal muscle mass, and vitamin D were adjusted and used as propensity variables.” in the limitation of the Discussion.

However, we have already described “and 33 (24.6%) were excluded due to mental health issues such as dementia, delirium, depression, and mental retardation.” in the Methods.

The statistical analysis section does not mention the ROC analysis. When comparing HSA values between the groups, consider using paired t tests to account for the matched design (see Austin 2007; A critical appraisal of propensity-score matching in the medical literature between 1996-2003).

Response: Thank you for your valuable comment. We have described “A receiver operator curve analysis (ROC) was also performed to identify the cutoff value for diagnosis of hip fracture using Hip structural analysis.” in the Methods.

In addition, we have described “To account for the matched design, we performed paired t tests. [24]” in the Methods.

In the discussion, address the variability in results by femur location (i.e. neck, shaft and IT).

When predicting fracture, is one more important than the other or should all three regions be taken into consideration?

Response: Thank you for your valuable comment. We absolutely agree with your consideration. Therefore, we have described “Fourth, we could not classify all types of hip fracture due to the limited number of samples in this study. Hip fractures include femur neck and intertrochanter, which can cause variability when interpreting the HSA values ​​of the femoral region. In future studies, it should be considered that factors such as fracture type, treatment for osteoporosis and duration of treatment can affect the thickness and width of the cortical bone around the femur. in the limitation section of the Discussion.

There was no text in the discussion comparing these values to other published data- would be nice to see whether they are similar – especially the hip axis length. I think that a next step worth mentioning is that these cut points should be tested in future studies to determine if they are as effective as BMD in predicting fracture. Also address that the current analysis pooled data from men and women when there may be important differences worth investigating in larger samples. 

Response: We have described “In previous study, Faulkner KG and McClung M et al. [36] reported that an automatic hip axis length measurement of 11.0 cm is associated with a twofold increase in hip fracture risk compared with a woman with an average hip axis length. In addition, they suggested that a hip axis length value of 11.6 cm increases hip fracture risk by a factor of 4 compared with a woman with a normal hip dimension. In this study, the same cut-off points were not calculated because that the different races analyzed in previous studies. However, the trend of increasing HAL in the fracture group was the same pattern as the previous study. Therefore, the cut-off value measured in this study will need to be validated on a larger sample cohort.” in Discussion.

Reviewer 2 Report

This is a well-written manuscript on an interesting topic. My major comments are related to the methods and validity of results.

Convenience sampling usually results in selection bias and limited generalizability, and even if we match for covariates both study groups should be representative of a respective population. In addition, all major confounders have to be included in the analysis. Further, the data have to be collected at the same time and age point. If the two groups were statistically similar on a limited number of confounders only, were not representative of a total population, and the time and age gap for the data collection was 10 years the validity of results might be compromised.

Regarding the HF group, 51/134 patients from one hospital were selected. The group consisted of milder patients as more severe ones were excluded. Was the group representative of all hip fracture patients? Did you assess and match for comorbidities that is a major confounder? And if you used the cross-sectional survey data would it have been possible to match for socioeconomic variables as well? Please explain and assess for impact. Regarding the sample size, you state that despite a small size the statistical problems were solved by matching with a group of 2000. What does this number stand for?

Selection of controls was also confusing. It was not clear how the 51 out of 1050 were selected and was the initial 1050 group representative of a total population or a population without previous hip fracture? What was the source of data for screening for a previous hip fracture among controls? It was not clear when the controls were assessed, in 2008 or in 2018. Was DXA performed twice? Were the 2008 or 2018 data used for matching? If 2008 the validity should be explained as the time gap between groups was 10 years. Please explain and assess for impact. And a minor comment: the non-HF group cannot be called a „normal patient“ group, please correct in the abstract.

It is not well understood why was the informed consent waived. Was the study part of KNHANES and were the HF patients also participating at KNHANES? Were the HF patients informed about the current study when they consented for KNHANES in 2008? The time gap was 10 years, and it is questionable if they consented for a current hip fracture study back then. Please explain and assess for impact.

Discussion.

It would be interesting to read why the propensity score method was chosen.

Minor comments:

Lines 51-2: „Due to BMD“? You probably mean that most fractures happen in patients with osteopenia.

In the abstract, you present the differences in values using a +- sign. Does it stand for SD? Please explain. And if you prefer using acronyms in the abstract please add a full wording.

Author Response

Reviewer 2

Comments and Suggestions for Authors

This is a well-written manuscript on an interesting topic. My major comments are related to the methods and validity of results.

Convenience sampling usually results in selection bias and limited generalizability, and even if we match for covariates both study groups should be representative of a respective population. In addition, all major confounders have to be included in the analysis. Further, the data have to be collected at the same time and age point. If the two groups were statistically similar on a limited number of confounders only, were not representative of a total population, and the time and age gap for the data collection was 10 years the validity of results might be compromised.

Response: We absolutely agree with your comment. We have described “Fifth, the historical data with time gap was used as a control. However, the hip structural analysis data for large number of healthy people were measured in the KNHANES study only in 2008, so this data was used in this study. Since there is still debate on the usefulness of hip structural analysis, this preliminary study was conducted to obtain additional evidence. Therefore, further studies are needed in larger sample cohorts.” in the limitation of the Discussion.

Regarding the HF group, 51/134 patients from one hospital were selected. The group consisted of milder patients as more severe ones were excluded. Was the group representative of all hip fracture patients?

Response: We absolutely agree with your comment. We have described “Finally, adjustment for all comorbidities could not be performed due to limitations in retrospective data analysis. Among the sociodemographic variables, economic and educational levels were different from those of the control group's national data-based survey. Therefore, direct statistical analysis could not be performed. However, gender and age were corrected for demographic variables that had the greatest impact on fractures. In disease and laboratory tests, osteoporosis, skeletal muscle mass, and vitamin D were adjusted and used as propensity variables. in the limitation of the Discussion.

Did you assess and match for comorbidities that is a major confounder? And if you used the cross-sectional survey data would it have been possible to match for socioeconomic variables as well? Please explain and assess for impact.

Response: We absolutely agree with your comment. We have described “Finally, adjustment for all comorbidities could not be performed due to limitations in retrospective data analysis. Among the sociodemographic variables, economic and educational levels were different from those of the control group's national data-based survey. Therefore, direct statistical analysis could not be performed. However, gender and age were corrected for demographic variables that had the greatest impact on fractures. In disease and laboratory tests, osteoporosis, skeletal muscle mass, and vitamin D were adjusted and used as propensity variables.” in the limitation of the Discussion.

Regarding the sample size, you state that despite a small size the statistical problems were solved by matching with a group of 2000. What does this number stand for?

Response: That sentence was our mistake. We have changed “However, statistical problems were solved by comparing the two groups through PSM matching in the control group of 2,000.” to “However, statistical problems were solved by comparing the two groups through propensity score matching.” in the Discussion

Selection of controls was also confusing. It was not clear how the 51 out of 1050 were selected and was the initial 1050 group representative of a total population or a population without previous hip fracture? What was the source of data for screening for a previous hip fracture among controls?

Response: This study selected people without hip fractures (self-reported) as a control from data representing the Korean population, not representative data without hip fractures in Korea.

It was not clear when the controls were assessed, in 2008 or in 2018. Was DXA performed twice? Were the 2008 or 2018 data used for matching? If 2008 the validity should be explained as the time gap between groups was 10 years. Please explain and assess for impact.

Response: We absolutely agree with your comment. We have described “Fifth, the historical data with time gap was used as a control. However, the hip structural analysis data for large number of healthy people were measured in the KNHANES study only in 2008, so this data was used in this study. Since there is still debate on the usefulness of hip structural analysis, this preliminary study was conducted to obtain additional evidence. Therefore, further studies are needed in larger sample cohorts.” in the limitation of the Discussion.

And a minor comment: the non-HF group cannot be called a „normal patient“ group, please correct in the abstract.

Response: According to your comment, we have added subheadings to describe the hip fracture and non-hip fracture cohorts and spell out all abbreviations in abstract.

It is not well understood why was the informed consent waived. Was the study part of KNHANES and were the HF patients also participating at KNHANES? Were the HF patients informed about the current study when they consented for KNHANES in 2008? The time gap was 10 years, and it is questionable if they consented for a current hip fracture study back then. Please explain and assess for impact.

Response: This KNHANES data is open to everyone. (https://knhanes.cdc.go.kr)

Discussion.

It would be interesting to read why the propensity score method was chosen.

Response: The number of patient groups was small (51), but the number of selected controls in the national data was relatively high (1050). Therefore, a similar propensity score was used to select 51 controls.

Minor comments:

Lines 51-2: „Due to BMD“? You probably mean that most fractures happen in patients with osteopenia.

Response: We have deleted “due to BMD” in the Introduction.

In the abstract, you present the differences in values using a +- sign. Does it stand for SD? Please explain. And if you prefer using acronyms in the abstract please add a full wording.

Response: According to your comment, we have added explain for SD in Abstract.

Round 2

Reviewer 2 Report

A number of remarks were not sufficiently addressed.

Line 262-3: ...the direct statistical analysis could not be performed because of different economic/educational status. What does the „direct statistical analysis“ stand for? Comparison of the groups? In line with that: you now mention in the limitations that the economic/educational status was different between the groups. In your analysis, you presented age and sex only. Did you have the data for socioeconomic variables as well? If yes, why not match for those?

What do you mean by „gender and age were corrected for demographic variables that had the greatest impact“?

For both groups, did you exclude individuals with previous fractures and how? How did you know that the cases and controls had no history of HF before?

Please define matching and adjustment. In the text, you use both as synonyms. Or did you use both?

It should be mentioned already in the abstract/methods section that the design was retrospective.

You still use „a normal patient“ wording in the text, at least twice. What does „normal patient“ stand for? Any patient who does not have HF? What if a patient has cancer, would that patient be normal or non-normal? Please correct the wording.

Author Response

Thank you for your valuable comments. We revised manuscript point by point according to the reviewer’s comments. In addition, we attached the certification of English editing service.

Line 262-3: ...the direct statistical analysis could not be performed because of different economic/educational status. What does the „direct statistical analysis“stand for? Comparison of the groups? In line with that: you now mention in the limitations that the economic/educational status was different between the groups. In your analysis, you presented age and sex only. Did you have the data for socioeconomic variables as well? If yes, why not match for those?

Response: To avoid confusion, we have changed “Therefore, direct statistical analysis could not be performed.’ to “Therefore, we could not perform statistical analysis after matching socioeconomic variables.” in the Discussion. This study was a retrospective study that did not match the socioeconomic variables measured in the control group. Because sociodemographic variables were not the same question. Therefore, we could not perform matching with socioeconomic variables.

What do you mean by „gender and age were corrected for demographic variables that had the greatest impact“?

Response: Among the demographic factors, these two variables including gender and age are the most important. Because this sentence is confusing to understand, we have deleted However, gender and age were corrected for demographic variables that had the greatest impact on fractures.” in the Discussion.

For both groups, did you exclude individuals with previous fractures and how? How did you know that the cases and controls had no history of HF before?

Response: We have described “In addition, control group excluded patients who were hospitalized due to accidents and fractures. in the Method.

Please define matching and adjustment. In the text, you use both as synonyms. Or did you use both?

Response: Thank you for your valuable comment. We have changed “adjustment” to “matching” in the manuscript.

It should be mentioned already in the abstract/methods section that the design was retrospective.

Response: According to your comment, we have changed “The purpose of this study was to compare the hip structural analysis (HSA) levels of normal patients with those in a hip fracture group.” to “The purpose of this retrospective study was to compare the hip structural analysis (HSA) levels of patients with those of a hip fracture group.” in Abstract and Introduction.

You still use „a normal patient“ wording in the text, at least twice. What does „normal patient“ stand for? Any patient who does not have HF? What if a patient has cancer, would that patient be normal or non-normal? Please correct the wording.

Response: To avoid confusion, we have deleted “normal” in the Manuscript.
